# UIP-AD: Learning Unified Intrinsic Prototypes for Multimodal Anomaly Detection

## Abstract

Multimodal anomaly detection, which combines appearance (RGB) and geometric (3D) data, is crucial for enhancing industrial inspection accuracy. However, prevailing fusion strategies, whether based on separate memory banks or direct feature-level integration, struggle to model the joint probability distribution of multimodal normality, leaving them vulnerable to cross-modal inconsistencies. In this paper, we address this by shifting the paradigm from feature-stitching to learning a unified, conceptual representation via Unified Intrinsic Prototypes (UIPs). Our framework dynamically extracts a single, compact set of these prototypes from a deeply-fused feature space to holistically represent the joint appearance-geometry distribution of a given sample. These powerful UIPs then guide a novel reconstruction stage, where parallel decoders are driven by a bidirectional attention mechanism to enforce cross-modal consistency. Anomalies that violate this consistency fail the reconstruction process and are exposed as large, localizable errors. Extensive experiments show that our framework establishes a new state-of-the-art on multiple challenging benchmarks, including MVTec 3D-AD, Real-IAD D³, and Eyecandies, validating the superiority of our approach and offering a robust, principled solution for 3D industrial inspection.

## 1 Introduction

Industrial automated production places unprecedentedly high demands on product quality inspection Cao et al. (2024). Unsupervised anomaly detection has become a key technology in this field due to its ability to operate without a large number of defect samples Zou et al. (2022); Wang et al. (2024); Bergmann et al. (2019). However, 2D image-based methods inherently struggle with defects defined by three-dimensional geometry or subtle surface topography. The popularization of 3D sensors has thus spurred significant interest in 3D multimodal anomaly detection, a paradigm that combines appearance from RGB images with geometry from 3D data to enhance inspection accuracy Bergmann et al. (2021); Chu et al. (2023b). The core principle is the natural complementarity between these modalities: RGB images excel at identifying surface-level defects, while 3D data provides robust, explicit information for capturing structural anomalies Costanzino et al. (2024a); Chu et al. (2023b).

However, effectively fusing these heterogeneous modalities remains an open and critical challenge. Existing paradigms suffer from fundamental limitations, as illustrated in our motivating example in Figure 1. Current 3D multimodal methods can be broadly categorized into two main paradigms. The first relies on **separate memory banks** Bergmann & Batzner (2023); Chu et al. (2023a), building distinct repositories for appearance and geometric features. This approach, however, is often inefficient due to large memory and computational overhead Costanzino et al. (2024a); Wang et al. (2023). More critically, it employs a shallow, score-level fusion which cannot model the deep consistency between modalities, rendering the models vulnerable to cross-modal inconsistencies where each modality appears normal in isolation but is anomalous in combination Chu et al. (2023b).

The second paradigm attempts to avoid large memory banks by using reconstruction or mapping-based frameworks Zhang et al. (2023b); Costanzino et al. (2024b); Cai et al. (2023); Zavrtanik et al. (2024). These methods, while often more efficient, typically resort to direct feature-level fusion—such as concatenation or addition. This approach, however, is prone to modality dominance, where the feature-rich RGB stream can easily overwhelm the often more subtle but structurally critical

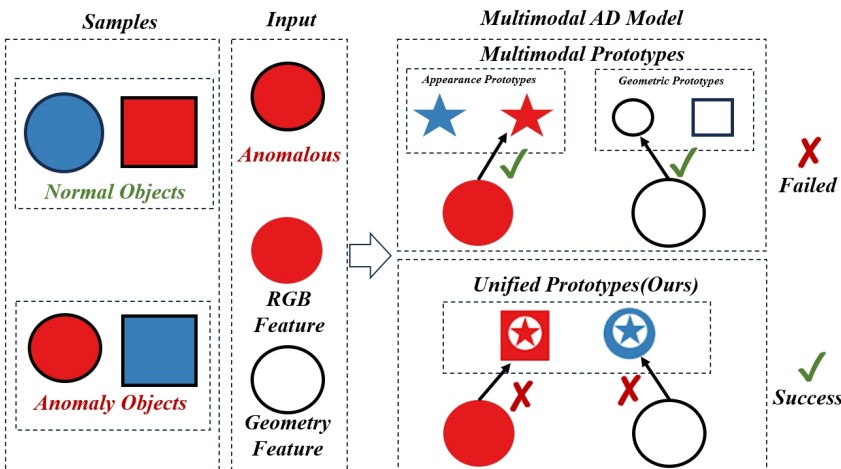

Figure 1: Motivation for UIP-AD. We illustrate a contextual anomaly where the 'normal world' (top left) consists of only blue circles and red squares. An anomalous 'red circle' (bottom left) fools methods that analyze modalities separately (top right), as both its red appearance and circular geometry are individually considered normal. In contrast, our method learns unified prototypes that represent the valid joint combinations of appearance and geometry (bottom right). Since no 'red circle' prototype was learned from the normal data, our method correctly identifies the input as anomalous.

geometric features. The challenge of designing an optimal fusion architecture is non-trivial and has itself become a recent research focus Long et al. (2025a;b). Consequently, a principled solution that avoids both the inefficiency of separate memory banks and the pitfalls of direct feature fusion is urgently needed.

We argue that the crux of the problem lies not in simply stitching features, but in learning a **unified, conceptual representation of normality**. This shifts the paradigm from feature-level to concept-level fusion. As depicted in Figure 1, we achieve this by learning a single, shared set of prototypes where each prototype itself is a joint appearance-geometry concept. This prototype set acts as a cross-modal universal language, holistically defining an object's normal state and thereby correctly identifying both geometric and cross-modal anomalies that fool other methods.

To realize this vision, we propose **UIP-AD**, an end-to-end framework that learns these **Unified Intrinsic Prototypes (UIPs)**. The key innovations of our framework are threefold:

1. We introduce the **Unified Cross-Modal Prototype Extraction (UCP)** module, which dynamically learns a compact set of UIPs from deeply-fused multimodal features.

2. We design a novel prototype-guided reconstruction stage where parallel decoders, while operating on modality-specific features, are guided by the single set of UIPs. The core of this stage is our **Cross-Modal Prototype Guided Attention (CPG)** block, which establishes a bidirectional interaction where prototypes guide feature reconstruction and features, in turn, refine the prototypes.

3. We develop a **Hierarchical Consistency Loss** that provides joint constraints on reconstruction quality ($L_s$), cross-modal semantic alignment ($L_a$), and prototype space diversity ($L_p$), enabling robust end-to-end optimization of the entire architecture.

Our comprehensive experiments show that UIP-AD sets a new state-of-the-art on MVTec 3D-AD, Real-IAD D³, and Eyecandies, validating the superiority of our unified prototype paradigm.

## 2 RELATED WORK

### 2.1 UNSUPERVISED IMAGE ANOMALY DETECTION

Unsupervised image anomaly detection is a mature field, with approaches generally falling into several categories. Reconstruction-based methods Schlegl et al. (2017); Zavrtanik et al. (2021); Zhang et al. (2023a) train generative models on normal data and identify anomalies as regions with high reconstruction error. Concurrently, student-teacher frameworks Rudolph et al. (2023) train a student network to mimic a teacher network's representations of normal data, flagging deviations as anomalous. However, feature-based methods have become dominant, operating in the latent space of powerful, pre-trained networks Cohen & Hoshen (2020); Defard et al. (2021). A prominent line of this work, including the influential PatchCore Roth et al. (2022), builds an external memory bank of normal features from the training set for nearest-neighbor comparison at test time. To address the rigidity and potential domain shift of external memory, a recent paradigm has shifted towards using intrinsic features extracted dynamically from the test sample itself. Works like INP-Former **?** pioneered this approach by defining normality based on a sample's own internal context. Our work is directly inspired by this advanced intrinsic paradigm, extending it to the multimodal domain.

### 2.2 MULTIMODAL ANOMALY DETECTION

The introduction of benchmarks like MVTec 3D-AD Bergmann & Sattler (2022) has accelerated research into combining RGB and 3D data for more robust anomaly detection. Early and prominent approaches extend 2D principles by employing separate memory banks for each modality, as seen in methods like BTF Xie et al. (2023), M3DM Bergmann & Batzner (2023), and ShapeGuided Chu et al. (2023a). While effective, this strategy incurs significant storage and computational overhead and performs only a shallow, score-level fusion, leaving it vulnerable to cross-modal inconsistencies. To address this inefficiency, a second wave of lightweight alternatives has emerged, including methods based on feature mapping Costanzino et al. (2024a) and efficient reconstruction Zhang et al. (2023b). These methods typically resort to direct feature-level fusion (e.g., concatenation), which can suffer from modality dominance where one modality's features overwhelm the other's. The limitations of both approaches have highlighted that the fusion strategy itself is a critical challenge. While works like 3D-ADNAS Long et al. (2025a) have begun exploring optimal fusion topologies, a principled framework for modeling the joint probability distribution of multimodal normality remains an open problem that we address in this paper.

## 3 PRELIMINARY: THE PROBLEM OF MULTIMODAL NORMALITY MODELING

Let $\mathcal{X}_{rgb} \subset \mathbb{R}^{H \times W \times 3}$ be the space of RGB images and $\mathcal{X}_{3d} \subset \mathbb{R}^{H \times W \times 3}$ be the space of corresponding pixel-aligned 3D data (e.g., depth maps). An unsupervised multimodal anomaly detection task aims to learn a model of normality from a training set $\mathcal{D}_{train} = \{(I_{rgb}, I_{3d})_k\}_{k=1}^{K}$ containing only normal samples. Given a test sample $(I_{rgb}^*, I_{3d}^*)$, the goal is to produce an anomaly map $S \in \mathbb{R}^{H \times W}$ where $S(i, j)$ is high if the spatial location $(i, j)$ is anomalous.

A highly effective approach is to model normality within a deep feature space. Let $\Phi_{rgb} : \mathcal{X}_{rgb} \to \mathcal{F}_{rgb}$ and $\Phi_{3d} : \mathcal{X}_{3d} \to \mathcal{F}_{3d}$ be feature extractors that map the input data into latent feature spaces. The core challenge then becomes how to properly model the joint probability distribution of normal features, $P(\Phi_{rgb}(I_{rgb}), \Phi_{3d}(I_{3d}))$.

**Limitation of Separate Prototype Spaces.** Many existing methods Bergmann & Batzner (2023); Chu et al. (2023a) implicitly model normality by learning separate sets of prototypes for each modality. Let $P_{rgb} = \{p_{rgb,m}\}_{m=1}^{M_1} \subset \mathcal{F}_{rgb}$ be a set of prototypes for the appearance modality and $P_{3d} = \{p_{3d,n}\}_{n=1}^{M_2} \subset \mathcal{F}_{3d}$ be a set for the geometric modality. Anomaly scoring for a patch at location $i$ is then typically a function of the distances to these separate prototype sets:

$$\text{Score}(i) = f(\min_m d(x_{rgb,i}, p_{rgb,m}), \min_n d(x_{3d,i}, p_{3d,n})), \tag{1}$$

where $x_{rgb,i}$ and $x_{3d,i}$ are the local features.

The fundamental limitation of this paradigm is its failure to model the **joint probability distribution** $P(x_{rgb}, x_{3d})$. Instead, it operates on the marginal distributions, $P(x_{rgb})$ and $P(x_{3d})$. This approach

assumes a sample is normal if its appearance is plausible and its geometry is plausible in isolation. However, this is an insufficient condition. A critical class of anomalies, which we term cross-modal inconsistencies, are those where the marginal probabilities $P(x_{rgb})$ and $P(x_{3d})$ are individually high, but the joint probability $P(x_{rgb}, x_{3d})$ is low. A red circle, for instance, is anomalous in a world composed only of blue circles and red squares. A separate prototype model would correctly identify "red" and "circle" as normal attributes, but would fail to detect that their combination is anomalous.

**The Case for a Unified Prototype Space.** Our work is motivated by the hypothesis that a more powerful representation of normality can only be learned by directly modeling the joint distribution. We propose to learn a single, **unified prototype set** $P_{uip} = \{p_{uip,k}\}_{k=1}^{M}$ within a shared, unified feature space $\mathcal{F}_{unified}$, which is produced by a fusion function $\Psi : (\mathcal{F}_{rgb}, \mathcal{F}_{3d}) \rightarrow \mathcal{F}_{unified}$. The core objective of our framework is to learn a set of prototypes $P_{uip}$ that effectively represents the true joint probability distribution of normal samples. Consequently, anomaly detection becomes a principled query of a fused test feature against this unified model of normality.

Formally, we can interpret the unified prototype space $\mathcal{F}_{unified}$ as an approximation to the joint distribution $P(x_{rgb}, x_{3d})$ via a finite mixture model. Each prototype $p_{uip,k} \in P_{uip}$ can be viewed as a latent "concept" representing a region of high probability mass. Specifically, given a fused feature vector $z = \Psi(x_{rgb}, x_{3d}) \in \mathcal{F}_{unified}$, we define an assignment weight

$$w_{ik} = \frac{\exp(\mathrm{sim}(z_i, p_{uip,k})/\tau)}{\sum_{j=1}^{M} \exp(\mathrm{sim}(z_i, p_{uip,j})/\tau)}, \tag{2}$$

where $w_{ik} \geq 0$ and $\sum_k w_{ik} = 1$. These coefficients $w_{ik}$ construct a categorical distribution over prototypes, which we interpret as a variational approximation of the posterior $P(k \mid z_i)$.

Under this view, the joint probability of a fused token $z_i$ can be approximated as

$$P(z_i) \approx \sum_{k=1}^{M} P(k)\, P(z_i \mid p_{uip,k}), \tag{3}$$

where $P(k)$ can be estimated by the empirical frequency of prototype usage, and $P(z_i \mid p_{uip,k})$ is approximated by the similarity-based soft assignment. Thus, our unified prototype set $P_{uip}$ effectively serves as an implicit, compact mixture model for the joint density $P(x_{rgb}, x_{3d})$ in the fused space.

In this formulation, anomaly detection reduces to querying whether a test feature $z$ has low likelihood under the learned mixture distribution, which in practice we realize through similarity- or reconstruction-based scoring using the same set of prototypes.

## 4 METHODOLOGY

Our proposed framework, UIP-AD, effectuates a paradigm shift from direct feature fusion to a more principled, concept-level fusion for 3D multimodal anomaly detection. As illustrated in Figure 2, the architecture is composed of three main stages: (1) Unified Cross-Modal Prototype Extraction (UCP), which learns a shared, intrinsic representation of normality from fused multimodal features; (2) Prototype-Guided Multimodal Reconstruction, which utilizes these prototypes to guide parallel decoders in rebuilding modality-specific features; and (3) a Hierarchical Consistency Loss that enables robust end-to-end optimization.

We denote the input RGB image as $I_{rgb} \in \mathbb{R}^{H \times W \times 3}$ and the corresponding depth map as $I_{3d} \in \mathbb{R}^{H \times W \times 3}$. A shared ViT encoder processes these inputs in parallel to produce multi-level feature sequences $E_{rgb} = \{E_{rgb,l}\}_{l=1}^{L}$ and $E_{3d} = \{E_{3d,l}\}_{l=1}^{L}$, where $L$ is the number of selected layers, and each feature map $E_{\cdot,l} \in \mathbb{R}^{B \times N \times D}$.

### 4.1 UNIFIED CROSS-MODAL PROTOTYPE EXTRACTION (UCP)

The UCP module is the cornerstone of our framework, designed to learn a single, shared set of prototypes. Each prototype in this set, a Unified Intrinsic Prototype (UIP), holistically represents a joint appearance-geometry concept, capturing the fundamental essence of normality.

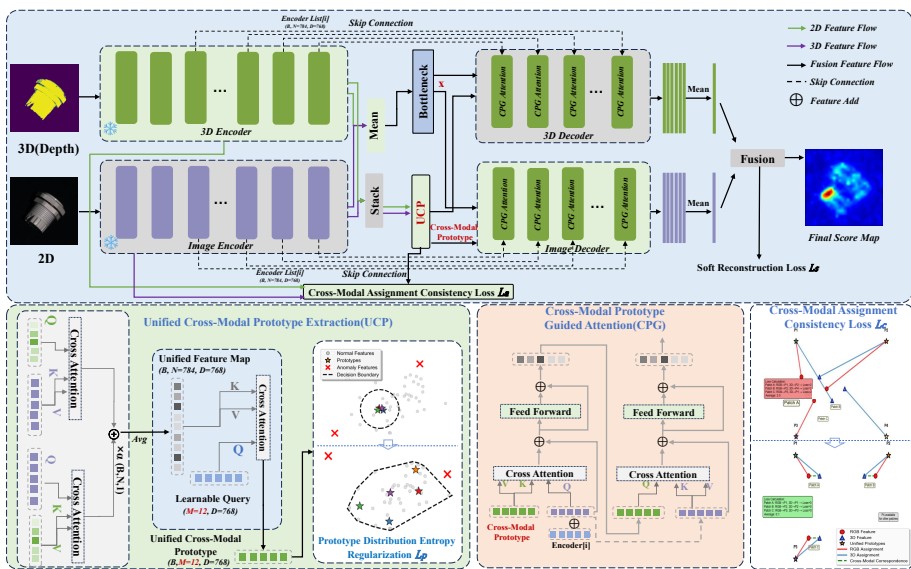

Figure 2: **Overview of the UIP-AD Framework.** Our framework detects anomalies through two primary stages during inference: (a) **Unified Prototype Extraction**. An input RGB image ($I_{\text{rgb}}$) and its 3D data ($I_{\text{depth}}$) are fed into a shared encoder to extract multimodal features ($E_{\text{rgb}}$, $E_{\text{3d}}$). After fusion, our **Unified Cross-Modal Prototype Extraction (UCP)** module dynamically aggregates these features to generate a single, compact set of Unified Intrinsic Prototypes ($P_{\text{uip}}$). These prototypes holistically represent the joint appearance-geometry concepts of a normal sample. (b) **Prototype-Guided Reconstruction**. The learned UIPs ($P_{\text{uip}}$) serve as a shared knowledge base to guide two parallel, modality-specific decoders in reconstructing the initial features.

The process begins with the fusion of multimodal features from the encoder. The multi-level feature sequences $E_{rgb}$ and $E_{3d}$ are first passed through separate Mean operations, which average the features across the $L$ layers to produce a single, representative feature map for each modality, denoted as $F_{rgb} \in \mathbb{R}^{B \times N \times D}$ and $F_{3d} \in \mathbb{R}^{B \times N \times D}$ respectively. These two feature maps are then concatenated and projected via a shallow MLP to form the final Unified Feature Map, $F_{unified} \in \mathbb{R}^{B \times N \times D}$. This map serves as the direct input for prototype generation.

The UCP module then dynamically generates $M$ UIPs. We initialize a set of $M$ learnable prototype tokens, $Q_{learn} \in \mathbb{R}^{M \times D}$, which act as the Query. The unified feature map $F_{unified}$ provides the Key and Value. Through a single layer of cross-attention followed by a feed-forward network (FFN), we obtain the final set of UIPs, $P_{uip} \in \mathbb{R}^{B \times M \times D}$. These prototypes are generated dynamically for each input sample and are thus intrinsic.

## 4.2 PROTOTYPE-GUIDED MULTIMODAL RECONSTRUCTION

The learned UIPs ($P_{uip}$) serve as a powerful, shared knowledge base to guide the reconstruction of both modalities. To ensure clear modality-specificity, our reconstruction stage utilizes two parallel, structurally identical decoders: an Image Decoder and a 3D Decoder. Critically, these decoders operate on their respective modality's features but are both guided by the *same* set of UIPs.

The initial inputs to this stage are the mean-averaged encoder feature maps, $F_{rgb}$ and $F_{3d}$. These are passed through a shared Bottleneck MLP before being fed into their corresponding decoders. Specifically, the Image Decoder processes the bottlenecked RGB features, while the 3D Decoder processes the bottlenecked 3D features. This separation ensures that each decoder focuses on reconstructing its own modality's characteristics, while the shared UIPs enforce cross-modal consistency.

The core of each decoder is a series of our novel Cross-Modal Prototype Guided Attention (CPG) blocks. Within each CPG block, a dynamic, bidirectional interaction occurs between the features being reconstructed and the shared prototypes. Let $F_{dec}^{(l-1)}$ (representing either RGB or 3D features)

and $P_{uip}^{(l-1)}$ be the feature map and prototypes entering the $l$-th CPG block. This interaction unfolds in two steps. First, in a prototype-guided feature update, the features are updated using the current prototypes as guidance. The feature map $F_{dec}^{(l-1)}$ acts as the Query, while the prototypes $P_{uip}^{(l-1)}$ serve as Key and Value.

$$F'_{dec} = F_{dec}^{(l-1)} + \text{Attention}(F_{dec}^{(l-1)}, P_{uip}^{(l-1)}, P_{uip}^{(l-1)}) \tag{4}$$

Second, in a feature-guided prototype refinement, the just-updated intermediate features $F'_{dec}$ are then used to refine the prototypes, with the prototypes $P_{uip}^{(l-1)}$ now acting as the Query.

$$P'_{uip} = P_{uip}^{(l-1)} + \text{Attention}(P_{uip}^{(l-1)}, F'_{dec}, F'_{dec}) \tag{5}$$

The final outputs of the block, $F_{dec}^{(l)}$ and $P_{uip}^{(l)}$, are obtained after passing $F'_{dec}$ and $P'_{uip}$ through their respective feed-forward networks. The refined prototypes $P_{uip}^{(l)}$ are then passed to the subsequent CPG block in both decoders, creating an iterative refinement cascade that is consistent across modalities.

## 4.3 HIERARCHICAL CONSISTENCY LOSS FUNCTION

To optimize our framework, we introduce a hierarchical loss, $L_{\text{total}}$, composed of three synergistic components:

$$L_{\text{total}} = L_s + \lambda_a L_a + \lambda_p L_p \tag{6}$$

**Reconstruction Loss** ($L_s$). This is the primary training objective, responsible for optimizing the **decoders** and the **bottleneck**. To logically connect the decoders to the prototypes, we define a soft reconstruction target using the prototypes themselves. First, for each original patch feature $\boldsymbol{x}_i \in F_{unified}$, we compute its soft reconstruction $\tilde{\boldsymbol{x}}_i$ as a weighted average of all prototypes $\boldsymbol{p}_j \in P_{uip}$, where the weights are determined by their similarity:

$$\tilde{\boldsymbol{x}}_i = \sum_{j=1}^{M} w_{ij} \boldsymbol{p}_j, \quad \text{where} \quad w_{ij} = \frac{\exp(\text{sim}(\boldsymbol{x}_i, \boldsymbol{p}_j)/\tau)}{\sum_{k=1}^{M} \exp(\text{sim}(\boldsymbol{x}_i, \boldsymbol{p}_k)/\tau)}. \tag{7}$$

This target $\tilde{\boldsymbol{x}}_i$ represents an idealized, normal version of the patch feature, synthesized from the learned concepts of normality. The decoders' role is to generate reconstructed features that match this idealized target. After the parallel decoders produce reconstructed features $\hat{F}_{rgb}$ and $\hat{F}_{3d}$, they are fused to create a unified reconstructed map $\hat{F}_{unified}$. The loss $L_s$ then penalizes the dissimilarity between the decoders' output and the soft reconstruction target:

$$L_s = \frac{1}{N} \sum_{i=1}^{N} (1 - \text{sim}(\hat{\boldsymbol{x}}_i, \tilde{\boldsymbol{x}}_i)), \tag{8}$$

where $\hat{\boldsymbol{x}}_i \in \hat{F}_{unified}$ is the fused output from the decoders for patch $i$. This process explicitly trains the decoders to map input features to their corresponding prototype-based normal representations.

**Cross-Modal Assignment Consistency Loss** ($L_a$). This loss acts as a semantic checksum to ensure the quality of the learned UIPs. It operates on the unfused encoder features ($F_{rgb}, F_{3d}$) and the final unified prototypes ($P_{uip}$). For each spatial patch $i$, we find the nearest prototype for its RGB feature $\boldsymbol{x}_{rgb,i}$ and its 3D feature $\boldsymbol{x}_{3d,i}$ by computing cosine similarity. As the prototypes $P_{uip}$ are generated from the fused features, they reside in a latent space that is inherently comparable to the modality-specific features from the shared ViT backbone, obviating the need for extra projection layers. Let $y_{rgb,i}$ and $y_{3d,i}$ be the one-hot vectors representing the nearest prototype indices. The loss penalizes assignment disagreement:

$$L_a = \frac{1}{N} \sum_{i=1}^{N} \|y_{rgb,i} - y_{3d,i}\|_1. \tag{9}$$

Its gradients primarily update the UCP module and the initial feature fusion stage.

**Prototype Distribution Entropy Regularization** ($L_p$). This loss regularizes the structure of the prototype space. It is calculated within the UCP module, acting on the Unified Feature Map

| Method | Year | Bagel | Cable Gland | Carrot | Cookie | Dowel | Foam | Peach | Potato | Rope | Tire | **Mean** |
|---|---|---|---|---|---|---|---|---|---|---|---|---|
| **Image-level AUROC (%)** | | | | | | | | | | | | |
| BTF Xie et al. (2023) | CVPR23 | 93.8 | 76.5 | 97.2 | 88.8 | 96.0 | 66.4 | 90.4 | 92.9 | 98.2 | 72.6 | 87.3 |
| EasyNet Zhang et al. (2023b) | MM23 | 99.1 | 99.8 | 91.8 | 96.8 | 94.5 | 94.8 | 90.5 | 80.7 | 99.4 | 79.3 | 92.6 |
| AST Cai et al. (2023) | WACV23 | 98.3 | 97.3 | 97.6 | 97.1 | 93.2 | 88.5 | 97.4 | **98.1** | 100.0 | 79.7 | 93.7 |
| M3DM Bergmann & Batzner (2023) | CVPR23 | 99.4 | 90.9 | 97.2 | 97.6 | 96.0 | 94.2 | 97.3 | 89.9 | 97.2 | 85.0 | 94.5 |
| ShapeGuided Chu et al. (2023a) | ICML23 | 98.6 | 89.4 | 98.3 | 99.1 | 97.6 | 85.7 | **99.0** | 96.5 | 96.0 | 86.9 | 94.7 |
| CFM Costanzino et al. (2024b) | CVPR2024 | 99.4 | 88.8 | **98.4** | **99.3** | 98.0 | 88.8 | 94.1 | 94.3 | 98.0 | 95.3 | 95.4 |
| 3D-ADNAS Long et al. (2025a) | AAAI2025 | **99.7** | **100.0** | 97.1 | 98.6 | 96.6 | 94.8 | 89.7 | 87.3 | **100.0** | 86.7 | 95.1 |
| UIP-AD (Ours) | - | 98.8 | 98.5 | 96.5 | 97.1 | **98.7** | 90.1 | 98.5 | 90.3 | 99.2 | **97.3** | **96.8** |
| **Pixel-level AUPRO (%)** | | | | | | | | | | | | |
| BTF Xie et al. (2023) | CVPR23 | 97.6 | 96.9 | 97.9 | 97.3 | 93.3 | 88.8 | 97.5 | 98.1 | 95.0 | 97.1 | 95.9 |
| EasyNet Zhang et al. (2023b) | MM23 | 83.9 | 86.4 | 95.1 | 61.8 | 82.8 | 83.6 | 94.2 | 88.9 | 91.1 | 52.8 | 82.1 |
| AST Cai et al. (2023) | WACV23 | 97.0 | 94.7 | 98.1 | 93.9 | 91.3 | 90.6 | 97.9 | 98.2 | 88.9 | 94.0 | 94.4 |
| M3DM Bergmann & Batzner (2023) | CVPR23 | 97.0 | 97.1 | 97.9 | 95.0 | 94.1 | 93.2 | 97.7 | 97.1 | 97.5 | 97.5 | 96.4 |
| ShapeGuided Chu et al. (2023a) | ICML23 | 98.1 | 97.3 | 98.2 | 97.1 | 96.2 | **97.8** | 98.1 | 98.3 | 97.4 | 97.5 | 97.6 |
| CFM Costanzino et al. (2024b) | CVPR2024 | 97.9 | 97.2 | 98.2 | 94.5 | 95.0 | 96.8 | 98.0 | 98.2 | 97.5 | 98.1 | 97.1 |
| 3D-ADNAS Long et al. (2025a) | AAAI2025 | - | - | - | - | - | - | - | - | - | - | - |
| UIP-AD (Ours) | - | **98.5** | **99.3** | **99.2** | **97.5** | **98.1** | 90.3 | **99.3** | **98.7** | **98.3** | **98.8** | **98.1** |

Table 1: Performance comparison on the MVTec 3D-AD dataset Bergmann et al. (2021). We report Image-level AUROC (%) and Pixel-level AUPRO (%). Best results are in **bold**, second-best are underlined.

($F_{unified}$) and the generated UIPs ($P_{uip}$). Let $q$ be the probability distribution of prototype assignments based on nearest-neighbor frequency. The loss is the negative entropy of this distribution:

$$L_p = \sum_{j=1}^{M} q_j \log(q_j + \epsilon), \qquad (10)$$

which encourages prototype diversity.

### 4.4 INFERENCE AND ANOMALY SCORING

At test time, the anomaly score for each patch is determined by its reconstruction error. This process begins with feature extraction and fusion, where a test sample ($I_{rgb}, I_{3d}$) is fed through the shared encoder to obtain $F_{rgb}$ and $F_{3d}$, which are then fused to produce the unified feature map $F_{unified}$. Next, the UCP module dynamically generates a set of intrinsic prototypes $P_{uip}$ from $F_{unified}$. Subsequently, the Image Decoder reconstructs $\hat{F}_{rgb}$ from $F_{rgb}$, and the 3D Decoder reconstructs $\hat{F}_{3d}$ from $F_{3d}$, with both decoders being guided by the same $P_{uip}$. The anomaly score is then calculated by fusing the reconstructed features to obtain $\hat{F}_{unified}$. The score $S_i$ for each patch $i$ is the dissimilarity between the original unified feature and its reconstruction from the decoders:

$$S_i = 1 - \text{sim}(\boldsymbol{x}_i, \hat{\boldsymbol{x}}_i), \qquad (11)$$

where $\boldsymbol{x}_i \in F_{unified}$ and $\hat{\boldsymbol{x}}_i \in \hat{F}_{unified}$. An anomalous patch, being inconsistent with the learned normal prototypes, will be poorly reconstructed, resulting in a high score $S_i$. Finally, for anomaly map generation, the patch-wise scores $\{S_i\}$ are reshaped to form a 2D heat map, which is then up-sampled to the original input resolution and smoothed with a Gaussian filter.

## 5 EXPERIMENTS

### 5.1 UNSUPERVISED SINGLE-CLASS ANOMALY DETECTION PERFORMANCE

In the unsupervised single-class setting, as shown in Table 1, Table 2, and Table 3, our proposed UIP-AD achieves state-of-the-art performance, outperforming all baselines on average across the three challenging benchmarks. On MVTec 3D-AD, UIP-AD obtains a mean Image-level AUROC of 96.8% and a mean Pixel-level AUPRO of 98.1% (values reported with optimal M=12).

The performance on Eyecandies and the large-scale Real-IAD dataset further validates the robustness and generalization of our approach. Notably, our method shows significant advantages in categories that require a deep understanding of the relationship between appearance and geometry, which we attribute to the principled modeling of their joint distribution via unified prototypes.

| Method | Candy Cane | Chocolate Cookie | Chocolate Praline | Confetto | Gummy Bear | Hazelnut Truffle | Licorice Sandwich | Lollipop | Marsh-mallow | Peppermint Candy | Mean |
|---|---|---|---|---|---|---|---|---|---|---|---|
| **Image-level AUROC (%)** | | | | | | | | | | | |
| AST Cai et al. (2023) | 57.4 | 74.7 | 74.7 | 88.9 | 59.6 | 61.7 | 81.6 | 84.1 | 98.7 | 98.7 | 78.0 |
| M3DM Bergmann & Batzner (2023) | 62.4 | 95.8 | 95.8 | **100.0** | 88.6 | 78.5 | 94.9 | 83.6 | **100.0** | **100.0** | 89.7 |
| CFM Costanzino et al. (2024b) | 68.0 | 93.1 | 95.2 | 88.0 | 86.5 | 78.2 | 91.7 | 84.0 | 99.8 | 96.2 | 88.1 |
| 3D-ADNAS Long et al. (2025a) | **89.6** | **100.0** | **97.0** | **100.0** | 82.7 | **88.2** | 93.1 | 95.0 | **100.0** | **100.0** | 94.6 |
| UIP-AD (Ours) | 85.2 | 98.7 | 96.5 | **100.0** | **95.6** | 86.9 | **95.4** | **95.8** | 99.8 | **100.0** | **95.7** |
| **Pixel-level AUPRO (%)** | | | | | | | | | | | |
| AST Cai et al. (2023) | 51.4 | 83.5 | 71.4 | 90.5 | 58.7 | 59.0 | 73.6 | 76.9 | 91.8 | 87.8 | 74.4 |
| M3DM Bergmann & Batzner (2023) | 90.6 | 92.3 | 80.3 | 98.3 | 85.5 | 68.8 | 88.0 | 90.6 | 96.6 | 95.5 | 88.2 |
| CFM Costanzino et al. (2024b) | 94.2 | 90.2 | 83.1 | 96.5 | **87.5** | 76.2 | 79.1 | 91.3 | 93.9 | 94.9 | 88.7 |
| 3D-ADNAS Long et al. (2025a) | **94.5** | 89.1 | 82.7 | 95.8 | 85.7 | 74.8 | **91.1** | 90.7 | 96.4 | 97.2 | 89.8 |
| UIP-AD (Ours) | 92.3 | **93.6** | **95.5** | **99.2** | 92.6 | 76.3 | 97.2 | 93.5 | **97.5** | **98.8** | **94.2** |

Table 2: Performance comparison on the Eyecandies dataset Costanzino et al. (2024a). We report Image-level AUROC (%) and Pixel-level AUPRO (%). Best results are in **bold**, second-best are underlined.

| Model | AST Cai et al. (2023) | PatchCore Roth et al. (2022) | M3DM Bergmann & Batzner (2023) | CFM Costanzino et al. (2024b) | D³M Zhu et al. (2025) | Ours |
|---|---|---|---|---|---|---|
| Metrics | IROC/PRO | IROC/PRO | IROC/PRO | IROC/PRO | IROC/PRO | IROC/PRO |
| audio_jack_socket | 0.860/0.510 | 0.926/0.536 | 0.981/0.711 | 0.983/0.701 | 0.983/0.714 | **1.000**/0.729 |
| common_mode_filter | **0.899**/0.576 | 0.523/**0.624** | 0.580/0.565 | 0.580/0.583 | 0.618/0.527 | 0.558/0.613 |
| connector_housing-female | 0.914/0.533 | 0.870/0.528 | 0.920/0.827 | 0.931/**0.889** | 0.931/0.849 | **1.000**/0.889 |
| crimp_st_cable_mount_box | 0.485/0.509 | 0.713/0.474 | 0.749/0.699 | 0.811/0.682 | 0.811/**0.733** | **0.986**/0.713 |
| dc_power_connector | **0.995**/0.566 | 0.720/0.646 | 0.715/0.802 | 0.922/0.788 | 0.922/**0.818** | 0.929/0.818 |
| ethernet_connector | **1.000**/0.749 | 0.947/0.860 | 0.983/0.850 | 0.996/0.828 | 0.996/0.889 | 0.848/**0.928** |
| ferrite_bead | 0.894/0.677 | 0.913/0.703 | 0.965/0.860 | 0.967/0.910 | 0.967/0.925 | **0.989**/**0.960** |
| fork_crimp_terminal | 0.595/0.599 | 0.780/0.613 | 0.780/0.548 | 0.780/0.821 | 0.819/0.582 | **0.828**/**0.889** |
| fuse_holder | 0.597/0.742 | 0.770/0.794 | 0.770/0.923 | 0.866/0.853 | 0.866/0.923 | **0.903**/**0.951** |
| headphone_jack_socket | 0.660/0.593 | 0.919/0.564 | 0.982/0.218 | **0.994**/0.865 | **0.994**/0.387 | 0.988/0.925 |
| humidity_sensor | 0.565/0.588 | 0.720/0.514 | 0.717/0.939 | 0.717/0.838 | 0.780/0.928 | **0.917**/**0.870** |
| knob_cap | 0.919/0.575 | 0.903/0.552 | 0.925/0.911 | **0.931**/0.924 | **0.931**/**0.960** | 0.917/0.824 |
| lattice_block_plug | 0.842/0.627 | 0.911/0.659 | 0.917/0.930 | 0.939/0.851 | 0.939/**0.950** | 0.953/0.821 |
| lego_pin_connector_plate | 0.847/0.638 | 0.662/0.625 | 0.681/0.792 | 0.891/0.832 | 0.891/0.870 | 0.868/**0.943** |
| lego_propeller | 0.471/0.547 | 0.540/0.516 | 0.530/0.796 | 0.739/**0.847** | 0.739/0.825 | **1.000**/0.729 |
| limit_switch | 0.804/0.689 | 0.822/0.725 | 0.863/0.828 | 0.925/0.881 | 0.925/0.821 | **0.960**/0.921 |
| miniature_lifting_motor | 0.766/0.672 | 0.948/0.677 | 0.975/**0.953** | 0.823/0.622 | 0.823/0.943 | **0.978**/0.582 |
| power_jack | 0.564/0.648 | 0.981/0.609 | **0.996**/0.721 | 0.973/0.322 | 0.973/**0.726** | 0.900/0.487 |
| purple_clay_pot | 0.635/0.606 | 0.921/0.606 | 0.944/**0.935** | **0.962**/0.917 | **0.962**/0.921 | 0.893/0.923 |
| telephone_spring_switch | **0.951**/0.588 | 0.827/0.588 | 0.856/0.877 | 0.934/0.856 | 0.934/**0.889** | 0.888/**0.889** |
| Avg | 0.763/0.612 | 0.816/0.621 | 0.841/0.784 | 0.860/0.791 | 0.890/0.812 | **0.918**/**0.819** |

Table 3: Performance comparison on the Real-IAD D³Zhu et al. (2025). We report Image-level AUROC and Pixel-level AUPRO (IROC/PRO). For each category, the best result per metric is in **bold** and the second-best is underlined.

## 5.2 QUALITATIVE ANALYSIS

Figure 3 presents a qualitative comparison of our method against several state-of-the-art approaches on representative samples from all three datasets. Across a variety of defect types, our UIP-AD consistently generates anomaly maps that are significantly sharper and more precisely localized to the ground truth regions. For instance, on the Eyecandies and Real-IAD D³ samples, other methods tend to produce diffuse heatmaps with considerable activation in normal regions, leading to potential false positives. In contrast, our method effectively suppresses the response on normal areas and provides a high-contrast localization of the defect. This is particularly evident on the MVTec 3D-AD examples, where our approach accurately identifies both textural and geometric anomalies with minimal background noise, highlighting the effectiveness of learning a unified prototype space.

## 5.3 COMPUTATIONAL EFFICIENCY ANALYSIS

To substantiate that learning intrinsic prototypes is more efficient than methods relying on large external memory banks, we conduct a computational cost analysis on the MVTec 3D-AD dataset. As detailed in Table 4, we evaluate inference throughput (Frames Per Second), peak GPU memory usage, and both detection (I-AUROC) and segmentation (P-AUPRO) performance. The results show UIP-AD demonstrates a superior balance of performance and efficiency. Compared to the resource-intensive memory-bank method M3DM, our framework is over **50×** **faster** and requires less than **10%** of the GPU memory, while also achieving higher accuracy on both metrics. This analysis confirms that our approach offers a practical and powerful solution for industrial inspection.

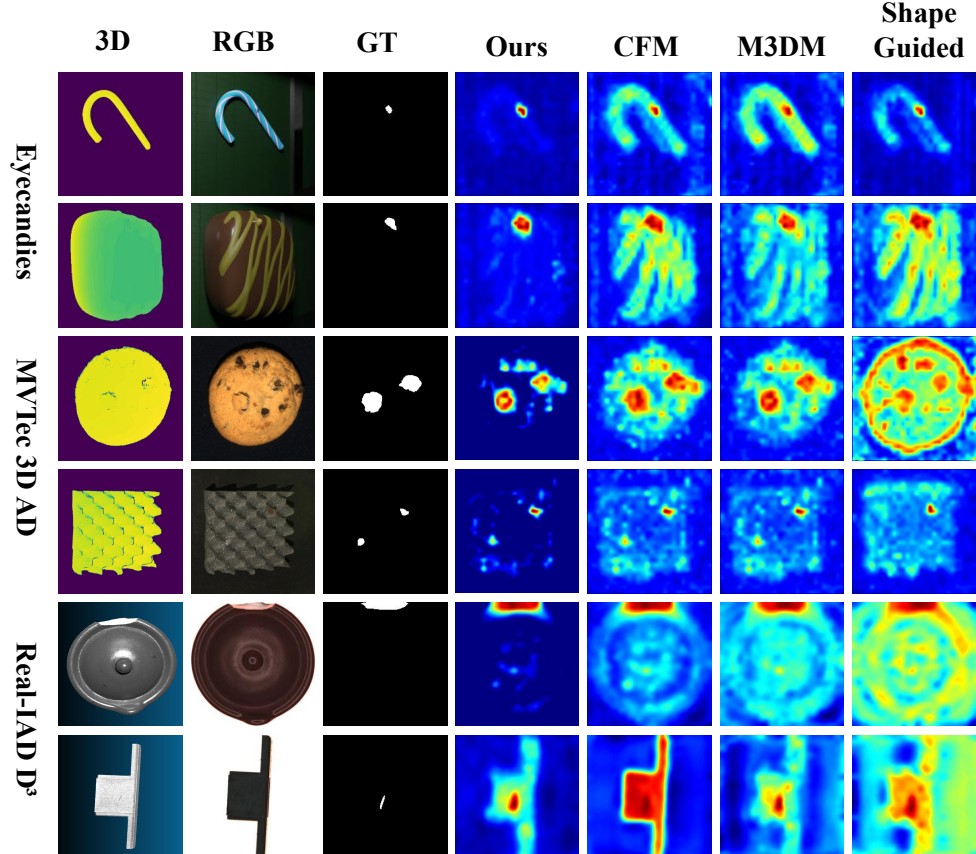

Figure 3: Qualitative comparison on samples from Eyecandies, MVTec 3D-AD, and Real-IAD D³. From top to bottom: Input RGB, 3D data, Ground Truth (GT), and anomaly maps from ShapeGuided, M3DM, CFM, and our UIP-AD. Our method produces significantly sharper and more accurate anomaly maps with fewer false positives.

| Method | Frame Rate (FPS) ↑ | Memory (MB) ↓ | I-AUROC (%) ↑ | P-AUPRO (%) ↑ |
|---|---|---|---|---|
| M3DM Bergmann & Batzner (2023) | 0.51 | 6526 | 94.5 | 96.4 |
| AST Cai et al. (2023) | 4.97 | 464 | 93.7 | 94.4 |
| CFM Costanzino et al. (2024b) | 21.7 | 437 | 95.4 | 97.1 |
| 3D-ADNAS Long et al. (2025a) | 24.7 | 269 | 95.1 | - |
| **UIP-AD (Ours)** | **28.5** | **382** | **96.8** | **98.1** |

Table 4: Computational efficiency analysis on MVTec 3D-AD. Our method provides a superior trade-off between speed, memory, and accuracy.

## 6 CONCLUSION

In this paper, we introduced UIP-AD, a novel framework for 3D multimodal anomaly detection that shifts the paradigm from feature-level stitching to concept-level fusion. By learning a single, compact set of Unified Intrinsic Prototypes (UIPs), our method holistically models the joint distribution of normal appearance and geometry. These powerful prototypes then guide a bidirectional reconstruction process to enforce cross-modal consistency, effectively exposing anomalies as large reconstruction errors. Extensive experiments on three challenging benchmarks validate that UIP-AD establishes a new state-of-the-art, demonstrating superior performance and efficiency. Our work provides a robust, principled, and effective solution for 3D industrial inspection and offers a strong new baseline for future research in multimodal fusion.

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

## A    APPENDIX

### A.1    ETHICS STATEMENT

This work adheres to the ICLR Code of Ethics. Our research focuses on unsupervised anomaly detection for industrial quality inspection, a domain with minimal risk of direct negative societal impact. The datasets used in this study, including MVTec 3D-AD, Real-IAD D³, and Eyecandies, are publicly available research benchmarks composed of inanimate industrial objects. Their use complies strictly with their respective licenses. Our methodology does not involve human subjects, animal experimentation, or personally identifiable information, and thus raises no privacy or security concerns. We have designed our framework for the general task of industrial inspection and are not aware of any inherent societal biases that could arise from its application. We are committed to the responsible and transparent advancement of machine learning.

## A.2 REPRODUCIBILITY STATEMENT

We are committed to ensuring the full reproducibility of our research. To this end, all source code for our models, training scripts, and evaluation procedures will be made publicly available in an anonymous repository upon publication. The experimental setup, including data preprocessing, model configurations, training hyperparameters, and hardware details, is described thoroughly in the main paper and the Appendix. We provide a detailed description of our core contributions, including the architecture of the **Unified Cross-Modal Prototype Extraction (UCP)** module, the implementation of the **Cross-Modal Prototype Guided Attention (CPG)** block with its bidirectional interaction, and the formulation of the **Hierarchical Consistency Loss**, to facilitate replication. Furthermore, all datasets used in our evaluation are public benchmarks, ensuring that our results can be consistently verified. We believe these measures provide a clear and complete roadmap for other researchers to reproduce our work.

## A.3 LLM USAGE

Large Language Models (LLMs) were utilized to assist in the writing and polishing of this manuscript. Specifically, we used an LLM for tasks such as improving sentence structure, checking for grammatical errors, and enhancing the overall clarity and flow of the text. The core scientific contributions, including the conceptualization of the Unified Intrinsic Prototype (UIP) paradigm, the design of the UIP-AD architecture, the formulation of the hierarchical loss function, the experimental methodology, and the analysis of the results, were developed exclusively by the authors. The role of the LLM was strictly limited to that of a writing aid. The authors have reviewed and edited all text and take full responsibility for the final content of the paper, ensuring its scientific accuracy and integrity.

## A.4 EXPERIMENTAL SETUP

**Datasets and Metrics.** We conduct extensive experiments on three challenging 3D multimodal anomaly detection benchmarks: **MVTec 3D-AD** Bergmann et al. (2021), **Real-IAD D³** Zhu et al. (2025), and **Eyecandies** Costanzino et al. (2024a). All three datasets provide registered pairs of RGB images and 3D data (depth maps or point clouds), which is essential for our multimodal approach. We follow the standard single-class training protocol, where a dedicated model is trained for each object category using only anomaly-free samples.

**Implementation Details.** All experiments are conducted on NVIDIA L40S GPUs using a PyTorch implementation. We employ a pre-trained and frozen DINOv2 Vision Transformer (ViT-Base/14) Oquab et al. (2023) as our shared backbone encoder. For data processing, input images (both RGB and depth) are resized to $448 \times 448$ pixels and then center-cropped to $392 \times 392$. Our model's encoder utilizes features from layers 2 through 9, which are organized into two groups for final feature comparison, following Luo et al. (2025). The Unified Cross-Modal Prototype Extraction (UCP) module is configured to learn a variable number of Unified Intrinsic Prototypes (UIPs), with our ablation study finding $M = 12$ to be optimal. The reconstruction stage utilizes two parallel, modality-specific decoders (one for image, one for 3D), each composed of 8 of our proposed Cross-Modal Prototype Guided Attention (CPG) blocks.

**Baselines.** We compare our method against a comprehensive set of state-of-the-art multimodal anomaly detection methods, including memory-bank-based approaches such as PatchCore Roth et al. (2022), BTF Xie et al. (2023), M3DM Bergmann & Batzner (2023), and ShapeGuided Chu et al. (2023a), as well as more lightweight frameworks like AST Cai et al. (2023), CFM Costanzino et al. (2024b), and EasyNet Zhang et al. (2023b). Furthermore, we include 3D-ADNAS Long et al. (2025a), which highlights the importance of fusion architecture.

## A.5 ABLATION STUDIES

To validate the effectiveness of the core components and design choices within our UIP-AD framework, we conduct two series of comprehensive ablation studies. The first study dissects the impact of our core architectural modules and hierarchical losses, while the second analyzes the model's sensitivity to key hyperparameters.

Figure 4: Qualitative results of our ablation study. From left to right: input RGB, 3D data, Ground Truth (GT), and the output of the Baseline, Baseline+UCP, Baseline+UCP+CPG, and our final model (Full UIP-AD).

| Configuration | Components | | | | | Performance |
|---|---|---|---|---|---|---|
| | Baseline | UCP | CPG | $L_a$ | $L_p$ | |
| *MVTec 3D-AD* | | | | | | |
| A: | ✓ | | | | | 93.1 / 94.2 |
| B: | ✓ | ✓ | | | | 94.3 / 95.7 |
| C: | ✓ | ✓ | ✓ | | | 95.5 / 96.9 |
| D: | ✓ | ✓ | ✓ | ✓ | | 96.1 / 97.5 |
| E: Full Model | ✓ | ✓ | ✓ | ✓ | ✓ | **96.8 / 98.1** |
| *Eyecandies* | | | | | | |
| A: | ✓ | | | | | 91.2 / 90.5 |
| B: | ✓ | ✓ | | | | 92.5 / 91.8 |
| C: | ✓ | ✓ | ✓ | | | 94.0 / 92.9 |
| D: | ✓ | ✓ | ✓ | ✓ | | 94.8 / 93.5 |
| E: Full Model | ✓ | ✓ | ✓ | ✓ | ✓ | **95.7 / 94.2** |
| *Real-IAD* | | | | | | |
| A: | ✓ | | | | | 88.2 / 79.3 |
| B: | ✓ | ✓ | | | | 89.4 / 80.0 |
| C: | ✓ | ✓ | ✓ | | | 90.7 / 80.9 |
| D: | ✓ | ✓ | ✓ | ✓ | | 91.2 / 81.3 |
| E: Full Model | ✓ | ✓ | ✓ | ✓ | ✓ | **91.8 / 81.9** |

Table 5: Ablation study on the core components of UIP-AD. We report mean I-AUROC / AUPRO (%).

### A.5.1 IMPACT OF CORE ARCHITECTURAL MODULES AND LOSSES

First, we evaluate the impact of our main architectural modules and hierarchical losses. As shown in Table 5, we start with a baseline model and incrementally add our key contributions, following a five-step evaluation (A-E). The progressive improvement in performance is also visualized in Figure 4. The configurations are as follows:

A. **Baseline**: A simple autoencoder with parallel decoders, supervised only by the reconstruction loss ($L_s$).

B. **+ UCP**: We introduce the Unified Cross-Modal Prototype Extraction module to guide the decoders.

C. **+ CPG**: The standard attention in the decoders is replaced with our Cross-Modal Prototype Guided Attention blocks, enabling bidirectional refinement.

**D.  + $L_a$:** We add the cross-modal assignment consistency loss to enforce semantic alignment.

**E.  + $L_p$ (Full Model):** We add the prototype distribution entropy loss. This represents our final UIP-AD model.

The results clearly demonstrate that each component provides a significant and consistent performance gain across all datasets. Moving from the baseline (**A**) to a prototype-guided approach (**B**) yields a significant gain, validating our core idea. Enabling bidirectional refinement with CPG blocks (**C**) further improves localization accuracy. Finally, the introduction of our hierarchical consistency losses, $L_a$ (**D**) and $L_p$ (**E**), provides the final increments in performance. The consistent gains across all three datasets confirm that all proposed components are integral and robust contributors.

| Num Prototypes (M) | MVTec 3D-AD | Eyecandies | Real-IAD |
|---|---|---|---|
| M = 6 | 96.5 / 97.8 | 95.4 / 93.9 | 91.5 / 81.6 |
| M = 8 | 96.7 / 98.0 | 95.6 / 94.1 | 91.7 / 81.8 |
| **M = 12 (Default)** | **96.8 / 98.1** | **95.7 / 94.2** | **91.8 / 81.9** |
| M = 16 | 96.6 / 97.9 | 95.5 / 94.0 | 91.6 / 81.7 |
| M = 24 | 96.5 / 97.8 | 95.3 / 93.8 | 91.5 / 81.6 |
| M = 32 | 96.4 / 97.6 | 95.2 / 93.7 | 91.4 / 81.5 |

Table 6: Ablation study on the number of prototypes ($M$). Performance is robust, with $M = 12$ being optimal. We report mean I-AUROC / AUPRO (%).

### A.5.2 SENSITIVITY TO THE NUMBER OF PROTOTYPES

In our second ablation study (Table 6), we analyze the model's sensitivity to the number of prototypes ($M$). We evaluate the full UIP-AD model with different values of $M$ on all three datasets.

**Analysis.** The results show that the performance of UIP-AD is highly robust to the number of prototypes. Across a wide range of values from $M = 6$ to $M = 32$, performance remains consistently high with only minor fluctuations. This demonstrates that our method is not overly sensitive to this hyperparameter. We observe that $M = 12$ yields a marginally better trade-off between representation capacity and redundancy, and thus we use it as the default setting for all other experiments.

