# OpenReview forum: "UIP-AD: Learning Unified Intrinsic Prototypes for Multimodal Anomaly Detection"
_ICLR.cc/2026/Conference — ICLR 2026 Conference Withdrawn Submission_

### Official Review · Reviewer_YhUR · 2025-10-26

**Soundness:** 2
**Presentation:** 2
**Contribution:** 4
**Rating:** 4
**Confidence:** 4

**Summary:**

The paper proposes a multimodal anomaly detection method (RGB + Depth) named UIP-AD. The method works is based on the inherent idea in INP-Former, which extracts features from the image and then reconstructs them based on those. To improve its performance in the RGBD domain, they propose unified prototypes (that is, not having separate RGB and D prototypes) to enable interaction between RGB and Depth data.

**Strengths:**

- Having unified prototypes makes sense on paper and achieves great results. The loss functions are also extremely well motivated and thought out.
- The model works at a very high speed and low memory consumption, making it suitable for actual scenarios

**Weaknesses:**

- There is no ablation study confirming some of the core ideas. Here are some high-level ablations I deem necessary: experiment with separate prototypes, the importance of each loss function, and the robustness to the number of prototypes
- There are some methods that outperform it on MVTec 3D: 3DSR (WACV’24) and TransFusion (ECCV’24 - results are on their Github). While I do not think these results undermine the contribution of the paper, I would recommend adding the results of these two methods to Table 1 for the sake of completeness.
- It would be interesting to see how the model performs with different encoders.
- It is unclear how well the model performs when only RGB or only Depth data is available.

**Questions:**

I have several questions. I have sorted them from most problematic to least problematic. If the first 2 questions are answered and they do not produce any conflicting results (the answers should also be added to the main paper), I am willing to raise my score.

1. What is the performance of the model when the model has separate prototypes?
2. How robust is the performance of the model when one of the loss functions is excluded (I am mostly interested in Cross-Modal Assignment Consistency Loss and Prototype Distribution Entropy Regularization).
3. How does UIP-AD perform in RGB-only or Depth-only scenarios?
4. How does UIP-AD work with other backbones?

---

### Official Review · Reviewer_6TJu · 2025-10-31

**Soundness:** 2
**Presentation:** 1
**Contribution:** 2
**Rating:** 2
**Confidence:** 4

**Summary:**

The paper introduces UIP-AD, a new framework for multi-modal anomaly detection. The authors point out that existing methods, which rely on separate memory banks or simple feature concatenation, struggle to effectively model the joint probability distribution of normal data. This makes them vulnerable to cross-modal inconsistencies. To address this, the paper proposes dynamically learning a set of unified and compact "Unified Intrinsic Prototypes" (UIPs) from a fused feature space. These prototypes holistically represent the valid joint concepts of appearance and geometry. The model then uses these UIPs to guide the reconstruction process. Experiments on three challenging benchmarks (MVTec 3D-AD, Real-IAD D³, and Eyecandies) show that UIP-AD achieves new state-of-the-art results.

**Strengths:**

+ The paper's motivation is very clear and direct. The problem of "cross-modal inconsistency" clearly calls for a unified representation of normality, and the subsequent design of the prototypes aligns well with this goal.
+ The proposed Cross-modal Prototype-Guided (CPG) attention module is well-designed. It creates a dynamic, bidirectional interaction between features and prototypes. This effectively enforces cross-modal consistency during reconstruction, which directly addresses the paper's core motivation.
+ The paper is evaluated on three popular datasets, demonstrating state-of-the-art performance in both image-level detection and pixel-level localization. The ablation studies also effectively demonstrate the contribution of each component.

**Weaknesses:**

+ Section 4.3 defines the loss as the difference between the reconstructed feature x̂_i and a soft reconstruction target x̃_i. This approach is quite unconventional and differs from standard practice. It is also inconsistent with how the score is calculated during inference. With this setup, it seems difficult for the model to converge to a state where the reconstruction error against the original feature is small. The authors need to provide a clearer justification for this choice of loss function.
+ The cross-modal assignment consistency loss forces both RGB and 3D features to map to the same prototype. This could potentially increase the number of prototypes required, which in turn would make the training process more difficult.
+ Nearly all the citations in Table 1 and the subsequent experiments are incorrect, some of the citations are not exist This is very strange. If the authors were familiar with this research area, they should have easily spotted these errors.

**Questions:**

Please see the **weakness** section.

---

### Official Review · Reviewer_jHAH · 2025-11-01

**Soundness:** 3
**Presentation:** 3
**Contribution:** 3
**Rating:** 6
**Confidence:** 5

**Summary:**

This paper proposes **UIP-AD**, a concept-level multimodal anomaly detection framework for industrial inspection that explicitly models the **joint distribution** of appearance (RGB) and geometry (3D) features using **Unified Intrinsic Prototypes (UIPs)**. Instead of stitching features or relying on large memory banks, the method (i) fuses RGB/3D embeddings, (ii) **extracts a compact set of joint prototypes** via a Unified Cross-Modal Prototype (UCP) module, and (iii) performs **bidirectional, prototype-conditioned reconstruction** through Cross-Modal Prototype-Guided attention. Experiments on **MVTec 3D-AD, RealIAD D3, and Eyecandies** show **state-of-the-art** image-/pixel-level performance with **substantial speed and memory gains** over strong baselines. Ablations, computational analyses, and qualitative visualizations support the core claims, and the approach is positioned as a shift from shallow fusion to **concept-/prototype-level joint modeling**.

**Strengths:**

- **Conceptual originality & clarity:** Moves multimodal fusion from shallow feature concatenation/memory banks to **joint prototype modeling** with an explicit probabilistic view (e.g., P(z_i)). The architecture (UCP + prototype-guided reconstruction) is coherent and well motivated.
- **Strong empirical quality & significance:** Across **three benchmarks**, the method achieves SOTA on both image- and pixel-level metrics and shows **dramatic efficiency gains** (e.g., vs. M3DM), which matters for deployment.
- **Ablations & diagnostics:** Extensive ablations on modules/losses and qualitative maps make a convincing case that each component contributes materially to performance.

**Weaknesses:**

- **Major**



  1. **Prototype count & expressivity:** The fixed number of prototypes M may underfit highly multimodal normals or rare subtypes. While **Table 6** shows robustness across a range of M, discuss scaling/selection strategies for richer distributions and potential adaptive mechanisms.
  2. **Evaluation on cross-modal inconsistencies:** Central claim includes robustness to joint-only anomalies, but **Tables 1–3**/**Figure 3** do not clearly isolate such cases. Add curated or synthetic **cross-modal outlier** tests to substantiate this aspect.
  3. **Loss target clarity:** In §4.3, the reconstruction loss L_s appears to compare reconstructed variants rather than original features. Clarify the objective/formula to avoid ambiguity and ensure standard reconstruction-to-original matching.
  4. **Missing latest baselines/surveys:** Recent relevant works (e.g., **Lin et al., 2024**; **Du et al., 2024**; **Zhang et al., 2024**) are not cited/compared. Incorporate them in Related Work and, if feasible, as baselines.
  5. **Limited theoretical depth:** The probabilistic interpretation is appealing but lacks analysis of approximation quality for the joint density P(x_{\text{rgb}}, x_{\text{3d}}), assignment optimality, convergence, or coverage guarantees.
  6. **No OOD/drift evaluation:** The paper does not test robustness under OOD, severe concept drift, or transfer to downstream tasks—important for industrial deployments.



- **Minor**

  7. **Colormap/legends in ablations:** **Figure 4**’s colormap and legends are occasionally hard to parse; add clearer labeling and include qualitative **failure cases**.
  8. **Prototype interpretability:** No visualization/audit of UIPs; risk of redundancy/collapse isn’t assessed. Provide diagnostics (e.g., nearest-realization visualizations, diversity metrics).
  9. **Notation/typos:** Occasional inconsistencies (e.g., “appearance-geometry” vs. “appearancegeometry”) and long equations spanning lines; tighten notation and indexing.

**Questions:**

- **Prototype number & adaptivity:** How does performance scale when M grows beyond the tested range? Could adaptive prototype counts (e.g., Bayesian nonparametrics, sparsity/merge-split heuristics) work here?
- **Loss target specification:** Does L_s compare decoder outputs to prototype-averaged features or to the original features? If the former, what are the implications for distributional coverage and anomaly boundary sharpness?
- **Cross-modal inconsistency evaluation:** Can you provide evaluations explicitly targeting joint-only anomalies (marginals normal, joint abnormal), beyond the illustrative **Figure 1**?
- **Prototype visualization/semantics:** Do UIPs correspond to interpretable physical states? Any evidence of prototype collapse or redundancy (e.g., overlap metrics, t-SNE of assignments, diversity regularizers)?
- **Handling OOD/concept drift:** What happens under severe drift or unseen combinations at test time? Any mechanism (e.g., confidence, abstention, adaptation) to mitigate false positives?

---

### Official Review · Reviewer_tb1D · 2025-11-01

**Soundness:** 3
**Presentation:** 2
**Contribution:** 2
**Rating:** 4
**Confidence:** 4

**Summary:**

This paper proposes **UIP-AD**, a prototype-based multimodal anomaly detection method that unifies RGB (appearance) and 3D (geometric) information under a shared concept representation. The main idea is to learn a set of Unified Intrinsic Prototypes (UIPs) that represent joint appearance–geometry normality. These prototypes are dynamically inferred from multimodal features through a Unified Cross-Modal Prototype (UCP) module and guide both RGB and 3D reconstruction decoders via Cross-Modal Prototype-Guided Attention (CPG). A hierarchical consistency loss regularizes reconstruction fidelity and prototype diversity. Experiments on MVTec 3D-AD, Eyecandies, and Real-IAD D$^3$ datasets show consistent improvements over existing multimodal AD baselines, e.g., AST, M3DM, CFM, 3D-ADNAS.

**Strengths:**

1. The “red circle” illustration effectively highlights cross-modal inconsistencies when RGB (appearance) and shape (geometry) features are processed independently. The argument for unified concept modeling across modalities is logical and intuitive.
2. The idea to switch from feature-level concatenation to prototype-level fusion is conceptually reasonable.
3. The proposed UIP-AD achieves state-of-the-art results across three multimodal benchmarks and demonstrates competitive runtime and memory efficiency.
4. The proposed method is well-detailed and appears to be reproducible. Detailed training setups, architecture configurations, and dataset splits are included in the appendix.
5. The ablations on layer selection, number of prototypes, and loss terms are thorough and informative.

**Weaknesses:**

1. While the “red circle” illustration adequately points out cross-modal inconsistencies, the paper does not convincingly establish that such inconsistencies play a decisive role in real-world anomaly detection tasks. If this observation is intended as the key motivation or conceptual punchline of UIP-AD, it raises concern that the qualitative results fail to include concrete examples where these inconsistencies lead to observable improvements over baseline methods. Without such evidence, the claimed advantage remains largely illustrative rather than empirically substantiated.
2. The proposed method is directly motivated by INP-Former (CVPR 2025), which introduced Intrinsic Normal Prototypes and an attention-based feature reconstruction framework for anomaly detection. UIP-AD extends this idea from single-modality (RGB) to multi-modality (RGB + 3D). The relatedness to INP-Former can be observed via the use of intrinsic prototypes and attention-based reconstruction. Hence, in the paper, it would be more appropriate to position  UIP-AD as “a multimodal generalization of INP-Former” , not as an independent paradigm.
3. The manuscript cites “INP-Former (?)”, indicating that the reference is missing from the bibliography. Given that INP-Former serves as the principal conceptual predecessor to the proposed method, the absence of a proper citation undermines the paper’s credibility and conveys an impression of incomplete attribution to prior work.
4. Despite claiming to “extend” the intrinsic-prototype paradigm, the paper does not include INP-Former as a baseline in any of the experimental tables. Without a numerical comparison (even as a unimodal variant), it is impossible to quantify the contribution of the multimodal extension.
5. The proposed method reuses established mechanisms such as prototype learning, cross-attention, and reconstruction losses, and applies them to multimodal inputs. The key novelty is structural integration rather than a new algorithmic principle.
6. The formulas in the paper appear to be mostly technically correct, but need to be further clarified. For example, the loss defined in Equation (9) involves two one-hot vectors, representing the "nearest prototype indices.”  How is the gradient of this loss computed, and why is it differentiable?
7. The claim that unified prototypes “capture multimodal intrinsic normal concepts” is not backed by visualization or analysis. Showing prototype activation maps or modality-specific attention responses would make this claim more convincing.

**Questions:**

The authors are suggested to respond to those raised in **Weaknesses.**

---

### Note · Authors · 2025-11-19

**Comment:**

Dear Area Chair and Reviewers,

We appreciate the reviewers' feedback, particularly the comments from Reviewer 6TJu regarding the bibliography. Upon re-examination, we have identified several inaccuracies in the citations and errors in the manuscript's presentation that occurred during the drafting process.

We deeply regret these oversights. To uphold the academic standards of ICLR and ensure the accuracy of our work, we have decided to withdraw the paper to conduct a comprehensive revision.

We sincerely apologize to the Area Chair and the Reviewers for any confusion caused and for the time spent reviewing our work.

Sincerely,

The Authors

**Withdrawal Confirmation:**

I have read and agree with the venue's withdrawal policy on behalf of myself and my co-authors.